# Modulator Combination Improves In Vitro the Microrheological Properties of the Airway Surface Liquid of Cystic Fibrosis Airway Epithelia

**DOI:** 10.3390/ijms231911396

**Published:** 2022-09-27

**Authors:** Alessandra Ludovico, Oscar Moran, Debora Baroni

**Affiliations:** Istituto di Biofisica, Consiglio Nazionale delle Ricerche (CNR), Via De Marini 6, 16149 Genova, Italy

**Keywords:** cystic fibrosis, correctors, bronchial epithelial cell culture, ion transport, pericilial mucus properties

## Abstract

Cystic fibrosis (CF) is a genetic disease caused by mutations in the cystic fibrosis transmembrane conductance regulator (CFTR) protein, a plasma membrane protein expressed on the apical surface of secretory epithelia of the airways. In the airways, defective or absent function of the CFTR protein determines abnormalities of chloride and bicarbonate secretion and, in general, of the transepithelial homeostasis that lead to alterations of airway surface liquid (ASL) composition and properties. The reduction of ASL volume impairs ciliary beating with the consequent accumulation of a sticky mucus. This situation prevents normal mucociliary clearance, favoring the survival and proliferation of bacteria and contributing to the genesis of the CF pulmonary disease. We explored the potential of some CFTR modulators, namely ivacaftor, tezacaftor, elexacaftor and their combination Kaftrio^TM^, capable of partially recovering the basic defects of the CFTR protein, to ameliorate the transepithelial fluid transport and the viscoelastic properties of the mucus when used singly or in combination. Primary human bronchial epithelial cells obtained from CF and non-CF patients were differentiated into a mucociliated epithelia in order to assess the effects of correctors tezacaftor, elexacaftor and their combination with potentiator ivacaftor on the key properties of ASL, such as fluid reabsorption, viscosity, protein content and pH. The treatment of airway epithelia bearing the deletion of a phenylalanine at position 508 (F508del) in the CFTR gene with tezacaftor and elexacaftor significantly improved the pericilial fluid composition, reducing the fluid reabsorption, correcting the ASL pH and reducing the viscosity of the mucus. Kaftrio^TM^ was more effective than single modulators in improving all the evaluated parameters, demonstrating once more that this combination recently approved for patients 6 years and older with cystic fibrosis who have at least one F508del mutation in the CFTR gene represents a valuable tool to defeat CF.

## 1. Introduction

Mutations in the cystic fibrosis transmembrane conductance regulator (CFTR) gene cause cystic fibrosis (CF), the most common life-threatening monogenic disease among Caucasians [1]. CFTR is an anion channel that transports chloride and bicarbonate across the apical plasma membrane of epithelial cells. CFTR gating is regulated by the binding of ATP to its intracellular binding domains and cAMP-dependent phosphorylation is mandatory for channel activity [2,3]. Its absence or dysfunction impairs the transepithelial flux of ions and fluid in airway cells. The respiratory disorder is the major cause of morbidity and mortality of CF patients. Most of the health problems in CF pulmonary disease are determined by the buildup of a viscous and tenacious mucus primarily induced by the post-secretional dysregulation of fluid and ion transport of the airway surface liquid (ASL) that coats the epithelial cells of the airways. Mucus accumulation constitutes the environment for recurrent infections and chronic inflammation leading to epithelial damage, tissue remodeling and progressive deterioration of lung function, ultimately climaxing in respiratory failure [4].

More than 2000 mutations in the CFTR gene have been described [5], of which at least 352 are disease-causing [6,7]. CFTR mutations have been grouped according to the molecular mechanism they disturb: protein synthesis (class I), maturation (class II), gating (class III), conductance (class IV), plasma membrane abundance (class V) and stability (class VI) [8,9]. Many mutations are pleiotropic, thus, they fit in more than one class [8]. The most prevalent CFTR mutation, the deletion of a phenylalanine at position 508 (F508del) [10,11], is primarily characterised by an incomplete folding (class II) [12,13,14] but also presents gating defects (class III) [15,16,17] and a reduced cell surface permanence (class VI) [12,18,19].

In recent years, a great deal of effort has been devoted to the identification of compounds, particularly potentiators of gating mutations and correctors of processing defects, able to recover the underlying defects of CFTR. This process culminated in the regulatory approval of Trikafta^TM^ (USA) or Kaftrio^TM^ (EU), the combination of correctors tezacaftor and elexacaftor with the potentiator ivacaftor, that has radically changed the health outcomes for many CF patients [20]. Indeed, modulator therapy has been shown to significantly improve sweat chloride, pulmonary function, body weight, and overall quality of life in patients with CF [21]. While the current research has provided unequivocal evidence that treatment with CFTR modulators leads to positive clinical outcomes, we do not yet truly understand their mechanism of action at the molecular level and provide evidence of the link between the pharmacological correction of mutant CFTR and the clinical benefits observed in patients. The elucidation of these aspects could provide useful information to further improve the efficacy of modulator therapy, especially in light of the extension of its use in patients with rare class II CFTR mutations.

ASL thickness depletion and mucus viscoelastic abnormalities have long been postulated to be the underlying causes of lung deterioration, inadequate host defence capability, lung deterioration and increased susceptibility to bacterial infections [22,23]. We have previously demonstrated that the pharmacological correction of F508del CFTR with lumacaftor significantly reduced fluid reabsorption and mucus viscosity in human bronchial epithelial cell (HBEC) monolayers [24]. Recently, a study by Birket and collaborators demonstrated that the treatment with the potentiator ivacaftor positively impacted the transport rate, viscosity and layer depth of the mucus recovered from the lungs of a rat model harbouring the ivacaftor-sensitive G551D (class III) mutation [25]. Similarly, in HBEC from subjects with G551D (class III) or F508del (class II) mutations, the rehydration of the ASL retrieved by treatment with CFTR modulators determined a decrease of mucus concentration, a relaxation of mucin network ultrastructure and a significant improvement of mucus clearance [26].

In this work, we aimed to demonstrate that CFTR modulators that compose the drug Kaftrio^TM^ (the potentiator ivacaftor and the correctors tezacaftor and elezacaftor) can recover some physical-chemical and visco-elastic properties of the ASL that lines the surface of the CF airway epithelium.

To achieve this goal, we analysed the effect of Kaftrio^TM^ and each of its components on the properties of the ASL recovered from monolayers formed by HBEC obtained from CF subjects [27]. We found that treatment of F508del-CFTR epithelia with the single correctors leads to the fluidification of the mucus, the alkalinisation of the ASL and increased fluid reabsorption, leading to values closer to those measured in non-CF control epithelia. This improvement was found to be greater with Kaftrio^TM^ treatment.

## 2. Results

### 2.1. Transepitelial Conductance

The conductance (G) values obtained in non-CF epithelia (14,732 ± 1267 µS/cm^2^, n = 10) were significantly higher than those obtained in CF epithelia (5727 ± 1299 µS/cm^2^, n = 10), consistently with the virtually null expression of CFTR in the apical membrane of mutant cells (Figure 1a). The effect of the different treatments on CF HBEC was assessed in terms of difference in conductance measurement before and after treatment (ΔG). As expected, the DMSO vehicle treatment was not significantly different from zero in either non-CF (ΔG = −338 ± 689 µS/cm^2^, n = 6) and F508del mutant CF HBEC (ΔG = 396 ± 325 µS/cm^2^, n = 6). In CF HBEC, treatment with elezacaftor or tezacaftor, or with the triple combination Kaftrio^TM^, significantly increased ΔG to 2387 ± 437 µS/cm^2^ (n = 9), 2996 ± 692 µS/cm^2^ (n = 9) and 2558 ± 464 µS/cm^2^ (n = 9), respectively. The virtual absence of CFTR in the apical membrane of CF epithelium is corroborated by the absence of any effect of the potentiator ivacaftor (ΔG = 1662 ± 737 µS/cm^2^, n = 9), whose ΔG value was not statistically different from that of the DMSO control (Figure 1b).

### 2.2. Transepithelial Fluid Reabsorption and Protein Content of Airway Surface Liquid (ASL)

The fluid reabsorption rate of HBEC non-CF epithelia incubated with DMSO for 48 h was 1.06 ± 0.02 µL h^−1^ cm^−2^ (n = 12), as shown by Figure 2. Consistently with the presence of a dehydrated ASL [28,29,30], fluid reabsorption of control homozygous F508del HBEC treated with the vehicle DMSO was significantly higher, with an average value of 1.67 ± 0.10 µL h^−1^ cm^−2^ (n = 7). Treatment with the potentiator ivacaftor did not significantly modify Jw of CF HBEC (1.58 ± 0.11 µL h^−1^ cm^−2^, n = 6), while incubation with correctors tezacaftor, elexacaftor and the combination tezacaftor + elexacaftor + ivacaftor decreased fluid reabsorption in HBEC CF epithelia to 1.27 ± 0.08 (n = 4), 1.23 ± 0.11 (n = 6), and 0.98 ± 0.13 (n = 9) µL h^−1^ cm^−2^, respectively (Figure 2). 

The protein concentration in the ASL of non-CF and CF HBEC was not significantly different in all examined conditions as shown by Table 1.

### 2.3. ASL Rheological Properties

The micro-rheological properties of the fluid recovered from the apical side of HBEC epithelia were analysed using the Multiple Particle Tracking (MPT) [24,31,32,33]. As shown by Figure 3a,c,d, the viscosity of the ASL of non-CF HBEC, calculated from the plot of the mean square displacement against the time interval (1.72 ± 0.13 cPoises, n = 11), was significantly lower than that of CF epithelia (5.07 ± 0.19 cPoises, n = 12). 

The viscosity of the ASL of CF HBEC epithelia treated with the potentiator ivacaftor (4.58 ± 0.17, cPoises, n = 12) was not different from that of DMSO-treated CF HBEC (Figure 3a–d). Conversely, treatment with tezacaftor, elexacaftor and tezacaftor + elexacaftor + ivacaftor reduced the viscosity of the ASL of F508del HBEC epithelia to 3.61 ± 0.06 cPoises (n = 10), 3.13 ± 0.08 cPoises (n = 11) and 2.62 ± 0.12 cPoises (n = 10), respectively (Figure 3a,e–g). Analogous results were obtained when comparing the path of a bead in the mucus samples of mutant-CFTR epithelia treated with 1 µM of ivacaftor, 5 µM of tezacaftor, 5 µM of elexacaftor and 1 µM of ivacaftor + 5 µM of tezacaftor + 5 µM of elexacaftor, respectively (insets of Figure 3b–g).

The elastic component of the ASL, expressed by the coefficient α, represents the non-linearity of the mean square displacement of the beads in the mucus samples. The α values of ASL from non-CF HBEC (0.69 ± 0.06, n = 11) was not statistically different from that of DMSO-treated CF HBEC (0.84 ± 0.05, n = 12). Also, treatment with CFTR modulators did not cause any significant change of α, that resulted 0.86 ± 0.02 (n = 12), 0.82 ± 0.05 (n = 10), 0.85 ± 0.05 (n = 12) and 0.73 ± 0.04 (n = 10), in ivacaftor, tezacaftor, elexacaftor, and tezacaftor + elexacaftor + ivacaftor CF HBEC mucus samples, respectively (Figure 3b–g).

### 2.4. ASL pH

The data presented in Figure 4 show that the pH of the ASL of non-CF HBEC (7.42 ± 0.06, n = 7) was significantly more alkaline than that of CF cells (7.03 ± 0.11, n = 5). In CF HBEC epithelia, the treatment with potentiator ivacaftor did not change the ASL pH (7.12 ± 0.07, n = 6). Conversely, the incubation with CFTR correctors significantly alkalinised the ASL pH (7.23 ± 0.03, n = 7; 7.38 ± 0.04, n = 7 and 7.42 ± 0.04, n = 7 for tezacaftor, elexacaftor and tezacaftor + elexacaftor + ivacaftor, respectively).

## 3. Discussion

ASL is a thin fluid layer that forms a continuous barrier between the lungs and the external environment, lining the surface of the airway epithelia. ASL mucus performs several important defence functions: it traps particulates and microorganisms; it constitutes a moving medium that can be pushed by the cilia toward the oropharynx; it is a physical barrier that reduces loss of fluid through the airways; and, finally, it acts as a transport medium for secreted substances such as enzymes, defensins, collectins, antiproteases and immunoglobulins [34]. To ensure its homeostasis, the composition of ASL is finely regulated thanks to the transport of ions and water through a variety of ion channels and transporters differentially distributed along the apical and basolateral membranes of epithelial cells. The airway epithelium is basically absorptive. The driving force for transcellular sodium reabsorption is provided by the Na^+/^K^+^-ATPase located along the basolateral membrane [35], while the main mediator of apical transport is the epithelial sodium channel (ENaC) found in the apical membrane. Most of the chloride secretion in human airways is provided by CFTR [36] and to a lesser extent by other channels, such as the Ca^2+^-activated chloride channels (CaCCs), including TMEM16A (ANO1) channels [37,38,39]. The transepithelial transport of water is passive, being generally driven by the osmotic gradient [40]. In CF, the equilibrium of the system is compromised by CFTR mutations that cause the diminution of the CFTR-dependent chloride secretion and the lack of the CFTR-mediated inhibition of ENaC activity [41], ultimately leading to a decrease of the ASL volume. ASL dehydration causes the collapse of the mucociliary clearance and reflects negatively on the innate immune system [41,42,43]. The lack of CFTR function also reduces the secretion of bicarbonate whose release is crucial for mucins post-secretional expansion [35,44,45,46,47,48]. 

Great advances in the pharmacological treatment of CF have been made since the discovery of CFTR modulators, mainly potentiators and correctors, which can recover the defects underlying the disease. The approval of the triple combination (Trikafta^TM^/Kaftrio^TM^) tezacaftor, elexacaftor and ivacaftor for the treatment of CF patients 6 years and older who have at least one F508del mutation represents a milestone in the fight against the disease. The use of this CFTR modulator formulation expands the cohort of patients eligible for treatment to the 90% of the CF population, demonstrating significant improvements in lung function and overall patient quality of life [49,50]. The adequate characterisation of the effect of CFTR modulators paves the way to the possibility to improve their efficacy and to define new formulations in order to expand the population of CF patients who can benefit from the pharmacological treatment of the disease.

This work is aimed to explore the impact of Kaftrio^TM^ and its individual components on key properties of the ASL from HBEC: fluid reabsorption, mucus micro-rheology and pH. To achieve this end, we used HBEC epithelia, which are pseudo-stratified, ciliated, produce mucus and express the main transport systems of in vivo airways epithelia [27]. It has been established that HBEC monolayer cultures grown under the ALI condition constitutes an excellent ex vivo model that closely mimics most of the characteristics of human airway epithelium [27,51,52,53].

It has been demonstrated that correctors tezacaftor and elexacaftor and their combination with potentiator ivacaftor significantly increase the CFTR-mediated ion transport in cells expressing F508del CFTR [54]. Our measurements in F508del HBEC preparations of the transepithelial electrical resistance (TEER), expressed as transepithelial conductance (G = 1/TEER, see Figure 1a,b), are consistent with an increase of the anionic transport across the membranes as also shown in isolated cells and epithelium models [55,56]. Hence, our observations on the electrical resistance let us argue that the improvement in the CFTR-mediated ion transport, correlates with corrector’s capability to recover F508del-CFTR protein abnormal maturation, membrane localisation and function, when administrated singly or in combination [56,57].

Our data showed that fluid reabsorption of CF epithelia was almost 1.6 times higher than that of non-CF epithelia, which is consistent with the decrease of the thickness of the ASL layer [29,30,58,59]. Differently from lumacaftor, the first corrector approved for the use of CF patients, that did not influence the fluid reabsorption in HBEC [24], treatment of CF HBEC epithelia with new generation CFTR correctors tezacaftor and elexacaftor and, with the triple combination, Trikafta^TM^/Kaftrio^TM^, reduced fluid reabsorption to values closer to those of non-CF-epithelia (Figure 2). The less reabsorption, the greater the ASL volume. This result should therefore positively influence the mucociliary clearance of CF epithelia as well as ASL antibacterial defense capability in vivo. We also noted that treatment with the potentiator ivacaftor did not change fluid reabsorption in CF epithelia. These data highlight once again that the activity of the fraction of F508del protein that manages to reach the plasma membrane [60] is not capable of producing a change in fluid secretion comparable to that of healthy HBEC even after stimulation with ivacaftor. 

Next, we focused our attention on the concentration of the proteins, almost mucins, collected from CF and non-CF epithelia. Aware that the measure of the concentration of mucins in the airway epithelium of CF patients is complex and could generate contradictory results [61,62,63,64], we used the Bradford assay to assess the protein content in the ASL samples recovered from the apical side of non-CF and CF monolayers treated with ivacaftor, tezacaftor, elexacaftor or their triple combination Kaftrio^TM^. Contrarily to previous works reporting that the concentration of mucins in the secretions from CF patients was higher than that of healthy patient secretions [65], we found that the protein concentration was similar in all CF and non-CF mucus samples (Table 1). Retrieved results are in agreement with those obtained by Finkbeiner and co-workers, who applied different biochemical and biomolecular techniques to quantify mucins secretion in human airway gland mucous primary cells [66]. This finding allowed us to hypothesise that in our epithelium model, the mucin content is independent from the presence of the CF pathological condition and also by the treatment with CFTR modulators. The impact of the treatment with CFTR modulators on the viscosity of the mucus secreted by HBEC was evaluated by MPT, a technique that permits to analyse the rheological properties of mucus from a relatively small amount of sample (≤8 µL, [24,31,32,33,67]). Similarly to what was observed with lumacaftor, administered alone or in combination with ivacaftor [24], the pharmacological correction by tezacaftor, elexacaftor and almost all by their combination with potentiator ivacaftor significantly improved the viscoelastic properties of the mucus (Figure 3). The triple combination of modulators was confirmed to be more effective than single CFTR modulators in decreasing mucus viscosity of CF cells. In CF epithelia, the potentiator ivacaftor had any effect on mucus viscosity, due to little or no F508del in the uncorrected cell membrane. This further confirms that CFTR channel function, especially bicarbonate secretion [52], plays a central role in regulating ASL mucus homeostasis.

Although the value and the role of ASL pH from CF subjects is still matter of discussion among researchers [68], it is generally accepted that CF airway secretions have a lower pH than the slightly alkaline mucus of normal airway, in partly due to impaired bicarbonate secretion and in partly to inflammation [69,70,71]. While our results could be biased toward slightly more alkaline values caused by the free diffusion of carbonic dioxide from the fluid, we observed significant differences in the pH of the ASL from CF and non-CF primary human airway epithelia, being more acidic than the ASL from homozygous F508del HBEC. Treatment with correctors resulted in a significant increase of the pH of the CF ASL, whose value in the case of administration of triple combination of modulators approached to that of non-CF ASL (Figure 4). The alkalinisation of pH after treatment with correctors could be linked to the increase of bicarbonate permeability of the corrector-rescued CFTR [47,67,72,73] and could translate in vivo into an amelioration of the innate defense of CF epithelia, as already demonstrated in CF-null pigs [74]. 

In conclusion, we have shown that correctors tezacaftor, elexacaftor and their combination with potentiator ivacaftor (Trikafta^TM^/Kaftrio^TM^) are able not only to recover some of the basic defects of F508del CFTR, such as plasma membrane expression (correctors) and transport activity (potentiator), but also to improve the microrheological properties of the mucus, influencing three features that participate in the regulation of ASL mucus homeostasis: hydration, viscosity and pH. Importantly, the combination of the new-generation correctors elexacaftor and tezacaftor with the potentiator ivacaftor increased the microrheological properties of ASL to a greater extent than each compound alone. As already mentioned, these results correlated well with the ability of the correctors to improve the processing, trafficking and function of the mutant CFTR observed in vitro [54,57,75]. In summary, our outcomes suggest that the administration of combinations of CFTR modulators capable to target different CFTR defects is the best strategy to address the CF pulmonary disease [49,76,77]. 

The effectiveness of the triple-combination elexacaftor, tezacaftor and ivacaftor to clinically improve lung function and respiratory-related quality of life of patients with at least one F508del copy has greatly impacted the life of the 90% of the CF patient population [55,78]. The results of this work provide further evidence of the effectiveness of Kaftrio^TM^. In fact, tezacaftor and elexacaftor, the two correctors that compose it, have been shown to significantly improve some key properties of the ASL that lines airway epithelial cells, such as fluid reabsorption, pH and viscosity. It remains to be determined whether Kaftrio^TM^ or other combinations of CFTR modulators will be able to recover the processing defects of other rare CFTR mutations. 

## 4. Materials and Methods

### 4.1. Chemicals

Culture media and supplements used in this article were those described in Gianotti and co-workers [27]. Ivacaftor (VX770), tezacaftor (VX-661) and elexacaftor (VX-445) were purchased from Selleck Chemicals (Munich, Germany). If not explicitly indicated in the text all other chemicals were provided by Merck (Milan, Italy). CFTR modulators were dissolved in DMSO to have a final concentration of the vehicle in the solutions in contact with the cells ≤ 0.1%.

### 4.2. Human Bronchial Epithelial Cell Culture

F508del/F508del and non-CF HBEC, from two non-CF and CF donors, were provided by the “Primary Cultures Service” of the Italian Cystic Fibrosis Research Foundation, isolated by following the procedures described elsewhere [27]. For use in these experiments, HBEC cells were thawed and plate cultured in a 1:1 mixture of LHC-9 and RPMI 1640 (LHC9-RPMI) media for two additional passages. To obtain differentiated epithelia, HBEC were seeded at high density on Transwell permeable supports (Corning, code 3379, New York, NY, USA) with bilateral addition of serum-free LHC9-RPMI medium. After 48 h the medium was replaced only on the basolateral side (air-liquid-interface condition, ALI) with DMEM/F12 (1:1) plus 2% New Zealand fetal bovine serum (Life Technologies, Monza, Italy), hormones and supplements. The cells were maintained in ALI condition for 4–5 weeks before performing the experiments in order to promote further differentiation of the epithelium. The complete maturation of epithelia was checked by measuring the transepithelial electrical resistance (TEER) (see below). Seven- to 8-day-old epithelia with TEER ≥ 1 KΩ∙cm^2^ and a potential difference ≈ −20 mV were considered as completely differentiated [27]. To achieve the highest effect on the parameters that we intended to assess, monolayers were treated for 48 h [79,80,81,82,83,84] with 80 µL of a solution containing (in mM): 150 NaCl, 5 KCl, 1.2 CaCl_2_, 0.5 MgCl_2_ and 0.1 HEPES (pH 7.4) and 1 µM of ivacaftor, 5 µM of tezacaftor, 5 µM of elexacaftor, and 1 µM of ivacaftor + 5 µM of tezacaftor + 5 µM of elexacaftor or 0.1% DMSO as control vehicle, administered on the apical side of the monolayers. 

### 4.3. Transepithelial Electrical Measurements

Transepithelial electrical resistance (TEER) was measured with 4 mm Ag|AgCl chopstick electrodes with an epithelial voltohmmeter EVOM2 (World Precision Instruments, Sarasota, FL, USA). To monitor the electrical properties of the treated epithelia, we measured the TEER immediately before the beginning and at the end (48 h) of the pharmacological treatment. The measurements were expressed in terms of conductance (G = 1/TEER; µS/cm^2^).

### 4.4. Measurement of Fluid Reabsorption and Protein Content 

At the end of incubation with CFTR modulators, the fluid lining the apical surface of each epithelium was carefully aspirated and collected. The volume of each sample was evaluated gravimetrically. The net flux of fluid across the epithelium, J_W_, was calculated as:J_w_ = (V_i_ − V_f_)/(A × t)(1)
where V_i_ and V_f_ are the initial and final apical fluid volumes, A is the area of the epithelium (for Transwell support: 0.33 cm^2^) and t is the time interval between the addition of V_i_ and recovering of the remaining fluid V_f_. The protein content of each sample was determined by using the Bradford assay. If not tested immediately, samples were frozen and stored at −80 °C up to three weeks.

### 4.5. Micro-Rheology

The micro-rheological properties of the fluid recovered from the apical side of epithelia was evaluated using the multiple particle tracking (MPT) method [24,27,30,31,32,33]. The method has been controlled by measuring the viscosity of glycerol solutions with known viscosity (see Appendix A). In MPT, the time course of the position of the beads in suspension inside the medium to be studied is recorded. A 25-µL aliquot of mucus and 1 µL of solution containing 200 nm diameter yellow/green fluorescent polystyrene carboxylated beads (λ_exc_ = 488 nm, λ_em_ = 505–515 nm; Life Technologies) were mixed, and a sample of 8 µL of mucus containing ≤100 beads/field was deposited between two glass cover slips, and the borders were sealed to avoid evaporation. After an equilibration at room temperature for 20 min, beads were focused on the mid-height of the sample to exclude beads that might be interacting with the cover slips with a 60× (N.A. = 1.42) objective connected to a CCD video camera. Images of 1280 × 960 pixels were captured at a rate of 5 frames/s. The trajectory of the Brownian motion of the beads was recorded using the Multitracker plug-in of ImageJ [85]. About 400 beads were tracked in four to eight fields per sample. The movement of the fluorescent beads within the mucus in a given time interval, τ, is described by its mean squared displacement, <msd>, from which it is possible to calculate the diffusion coefficient D_0_, according to the equation:<msd(τ)> = 4D_0_ τ^α^
(2)
where α is the elastic contribution of the fluid whose value is 0 < α ≤ 1. The viscosity, η, is calculated from the Stokes–Einstein equation, as:η = k_B_ T/6π D_0_ r(3)
where k_B_ is the Boltzmann constant, T is the absolute temperature and r is the radius of the microspheres.

### 4.6. pH Measurement

After 48 h of incubation with CFTR modulators, mucus samples (≥40 µL) were quickly collected from the apical surface of each epithelium and their pH was immediately measured (within 2 min) using a microelectrode (SevenCompact-Mettler Toledo, Novate Milanese, Italy). The measurements were repeated at least six-fold.

### 4.7. Statistics

Analysis of variance was applied to verify that data obtained from the two non-CF and CF donors had the same fluctuation. Once verified that there was no difference among intra- and inter-group replicates, the replicates under each condition (5 to 12) were pooled as a single population. Data are shown as mean ± standard error of the mean (sem). All statistical and MPT analysis was done with IgorPro 9 software (Lake Oswego, OR, USA). The Kruskal–Wallis non-parametric analysis of variance followed by Dunn’s post hoc test were used to compare data sets. A value of *p* < 0.05 was considered statistically significant.

## Figures and Tables

**Figure 1 ijms-23-11396-f001:**
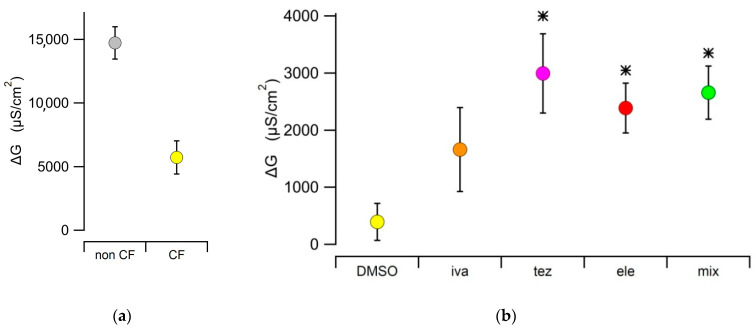
(**a**) Transepithelial conductance, G, in HBEC epithelial monolayers expressing WT (non-CF) and F508del mutant CFTR (CF). Measurements (n = 10 for both conditions) were obtained after treatment with DMSO as vehicle. (**b**) Difference between transepithelial conductances measured before and after the different treatments (∆G), measured in epithelia expressing the F508del mutation. (DMSO: 0.1% DMSO (vehicle control), iva: 1 µM of ivacaftor, tez: 5 µM of tezacaftor, ele: 5 µM of elexacaftor, mix: 1 µM of ivacaftor + 5 µM of tezacaftor + 5 µM of elexacaftor). The sample size, n, was comprised between 6 ≤ n ≤ 9. Asterisks indicate that the ∆G value is significantly different from that measured in CF HBEC treated with the DMSO vehicle.

**Figure 2 ijms-23-11396-f002:**
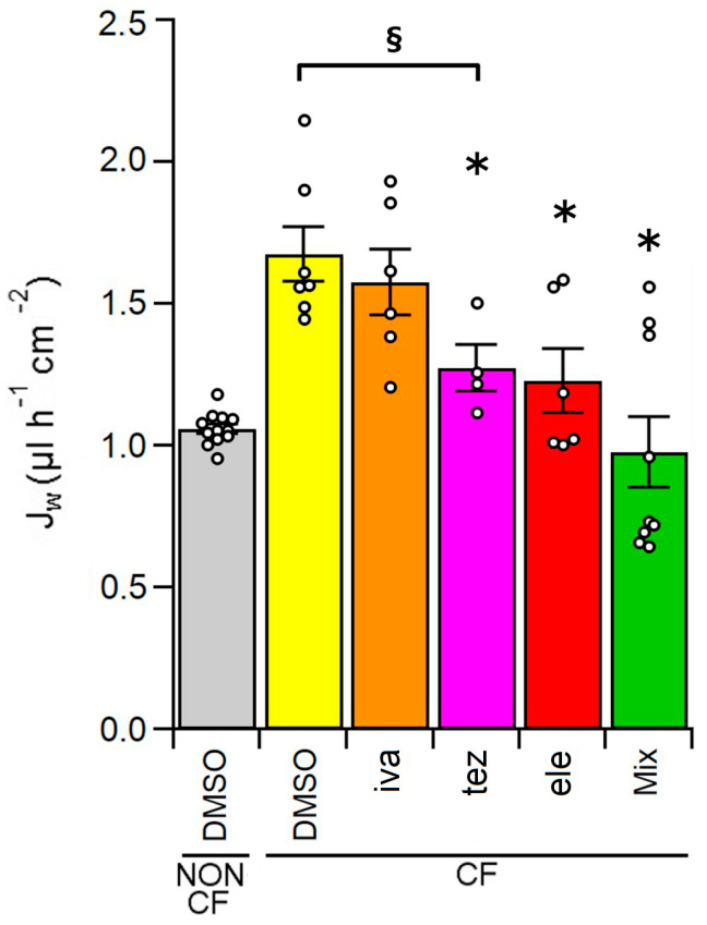
Fluid reabsorption, J_W_, of HBEC monolayers, cultured on Transwell permeable supports, from non-CF cells treated with 0.1% DMSO (vehicle control) and CF HBEC treated with 0.1% DMSO (vehicle control), 1 µM of ivacaftor (iva), 5 µM of tezacaftor (tez), 5 µM of elexacaftor (ele), 1 µM of ivacaftor + 5 µM of tezacaftor + 5 µM of elexacaftor (Mix) for 48 h. Fluid reabsorption of each sample was evaluated gravimetrically as described in the Section 4. The white circles represent the values of each individual measure (6 ≤ n ≤ 12). The section mark (§) indicates data that are statistically different from non-CF control cells, while asterisks (*) indicate a statistical significance versus control CF cells.

**Figure 3 ijms-23-11396-f003:**
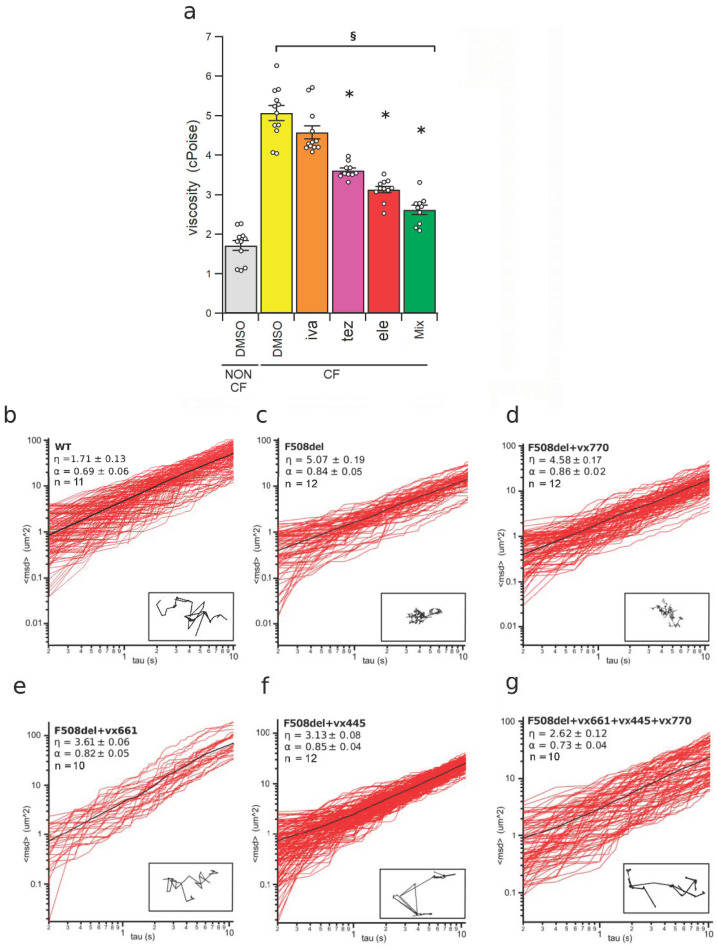
Micro-rheology of the airway surface liquid (ASL) recovered from HBEC monolayers measured by the multiple particle technique (MPT). (**a**) Viscosity of the ASL collected from the apical side of non-CF and CF epithelia. Non-CF HBEC epithelia were incubated with a solution containing 0.1% DMSO (vehicle control) while CF HBEC epithelia were treated with 0.1% DMSO (vehicle control), 1 µM of ivacaftor (iva), 5 µM of tezacaftor (tez), 5 µM of elexacaftor (ele), 1 µM of ivacaftor + 5 µM of tezacaftor + 5 µM of elexacaftor (mix), respectively, administered on the apical side of the monolayers. The white circles in (**a**) represent the values of each single measure (10 ≤ n ≤ 12). The section mark (§) indicates that data are statistically different from non-CF control cells, while asterisks (*) indicate a statistical significance versus control CF cells. (**b**–**g**) Plots of the square displacement, <msd> against the time interval of ≤100 beads in the ASL samples from non-CF HBEC (WT) treated with DMSO (**b**) and from F508del mutant CF HBEC (F508del) treated with DMSO (**c**), ivacaftor (VX770) (**d**), tezacaftor (VX661) (**e**), elexacaftor (VX445) (**f**), and ivacaftor + tezacaftor + elexacaftor (VX661 + VX445 + VX770) (**g**). The average <msd> is shown as a black solid line. The values of viscosity, η, the elastic modulus, α, and the sample size, n, are indicated in each panel. The insets in panels (**b**–**g**) show the trajectory path of a bead recorded in each ASL sample.

**Figure 4 ijms-23-11396-f004:**
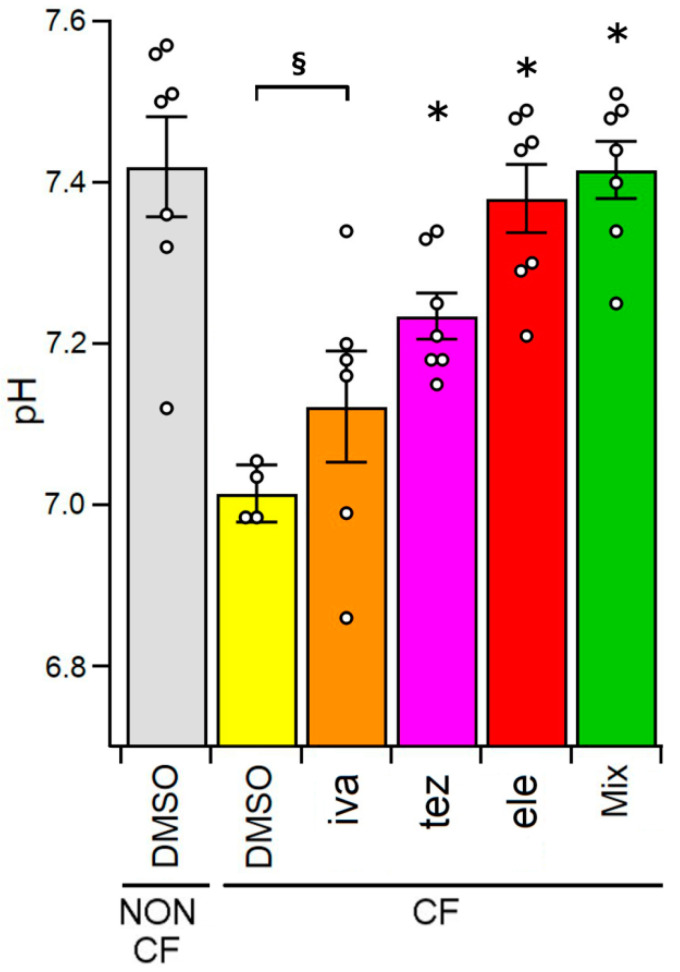
ASL pH measured from non-CF HBEC epithelia treated with 0.1% DMSO (vehicle control) and from CF HBEC epithelia incubated with 0.1% DMSO (vehicle control), 1 µM of ivacaftor (iva), 5 µM of tezacaftor (tez), 5 µM of elexacaftor (ele), and 1 µM of ivacaftor + 5 µM of tezacaftor + 5 µM of elexacaftor (mix), respectively. The white circles represent the values of each single measure. The section mark (§) indicates that data are statistically different from non-CF control cells, while asterisks (*) indicate a statistical significance compared to control CF cells.

**Table 1 ijms-23-11396-t001:** Protein concentration in ASL samples collected from the apical side of non-CF treated with 0.1 % DMSO (vehicle control) and CF HBEC treated with 0.1 % DMSO (vehicle control) or 1 µM of ivacaftor (iva), 5 µM of tezacaftor (tez), 5 µM of elexacaftor (ele) and 1 µM of ivacafotor + 5 µM of tezacaftor + 5 µM of elexacaftor (Mix) for 48 h. Protein concentration (µg/mL) was measured with the Bradford assay. Data are expressed as mean ± sem (standard error of the mean). n indicates the sample size.

	Non-CF	CF				
	DMSO	DMSO	iva	tez	ele	mix
Mean ± sem (µg/mL)	1.08 ± 0.03	1.18 ± 0.23	1.22 ± 0.21	1.12 ± 0.09	1.09 ± 0.10	1.00 ± 0.07
n	12	7	7	7	7	7

## Data Availability

The datasets used and/or analysed during the current study are available from the corresponding author on reasonable request.

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
