# Peer review of "Modulator Combination Improves In Vitro the Microrheological Properties of the Airway Surface Liquid of Cystic Fibrosis Airway Epithelia"

_ijms, 2022, doi:10.3390/ijms231911396_

Round 1

Reviewer 1 Report

The paper by A. Ludovico et al. investigates the effects of correctors tezacaftor and elexacaftor and their  combination with potentiator ivacaftor on the properties of airway surface liquid (fluid reabsorption,  viscosity, protein content and pH) in differentiated primary  human bronchial epithelial cells obtained from CF and non-CF patients. The authors demonstrate that the treatment ameliorate the transepithelial fluid  transport and the viscoelastic properties of the mucus when the correctors are used singly or in combination.

In general, the research design is appropriate with adequate description of the methods and presentation of the results, probably a final graphical scheme would help to better recapitulate the overal effects of the molecules used in the experiments. Anyway, I have some questions to be adressed before the acceptance of the paper.

I wonder why the treatment with the CFTR inhibitor ppq102 does not affect the viscosity of ASL in non-CF cells (though reducing significantly pH), since the CFTR is important also for bicarbonate secretion that in turn promotes the relaxation of mucin network. Does ppq102 affect the overall bicarbonate secretion? In my opinion this should be briefly discussed.

 I also wonder why the assays were performed after 48 hours treatment. Have you investigated the effects of the molecules in shorter times than 48 hours (i.e. 24 hours?) or in longer times to study the persistence of the effect (i.e 72 hours)? In my opinion, this issue should be discussed.

Author Response

please, see the attachment

Reviewer 2 Report

my main comments on this paper were a severe lack of attention to details and poor experimental design (or at least thorough explanation of the methods). Critical details are missing to judge the validity of the results.
If the authors apply CFTR modulator treatment via apical fluid for 48h and measure mucus rheology, it is impossible to determine the true value of elastic and viscous moduli. Long hydration will affect the viscoelastic properties of mucus, as well as activate or inhibit different ion channels to regulate the excess fluid on the cell surfaces. In addition, media left on the cell surface will change pH in an incubator, which is known to affect mucus rheology.
Authors used ppq102 to inhibit CFTR but had no effects on non-CF cultures (except maybe slightly on pH) and vaguely mention the lack of alteration of ENaC in the discussion to explain the lack of effect but this is not convincing. CFTR mutations only affect CFTR and as shown in their own data, CFTR dysfunction affects water absorption and mucus rheology. Regarding the use of ppq on CF cells, authors failed to mention if the inhibitor was used after modulator treatment, which would indeed somewhat demonstrate that the response they obtained was due to CFTR correction, but if ppq was used on untreated CF cells then there is no point to that experiment. Again, no explanation was given on this specific experimental design. Measuring pH in a non-controlled CO2 and temperature chamber will affect pH within milliseconds. Hence, another set of data that is questionable.
As for the write up of the manuscript, proofreading should have been done before submitting the paper. There is an entire paragraph that is repeated (line 109), reference to figures is sometimes missing in the results section (e.g., protein concentration), description/name of the assay is not specified in Figure 1, description of the rectangle window is not provided in figure legend for figure 2. Figure S1 is mentioned in discussion but not described in results section.
There are a lot of minor comments but these are the main points.

Author Response

please, see attachment

Reviewer 3 Report

The authors discuss a study regarding the superior effect of triple combination therapy (VX-661, VX-445, VX-770) on improving microrheological properties of airway surface liquid in patients with CF. They utilized primary human bronchial epithelial cells derived from non-CF and CF patients and established polarized monolayers under ALI conditions to model in vivo properties of airway epithelial cells. The study used established methods described in the literature to measure surface liquid reabsorption and Multiple Particle Tracking to compare micro-rheological properties of ASL from CF and non-CF monolayers. Additionally, they extend this method to calculate and compare the viscosity of the studied ASLs. The study's findings agree with the literature and provide evidence of the superior outcome of treatment with the triple combination of VX drugs compared to double (VX-661+VX770) or monotherapy (VX-661).

Minor comments:

1)      In Lines 138-142, the authors state that “ASL mucus performs several important defence functions: it traps particulates and microorganisms; it constitutes a moving medium that can be pushed by the cilia toward the oropharynx; it is an impermeable layer that reduces loss of fluid through the airways; and, finally, it acts as a transport medium for secreted substances such as enzymes, defensins, collectins, antiproteases, and immunoglobulins.”

I am arguing with the statement that the ASL is an impermeable layer. It is permeable for drugs and even for viruses and bacteria (for example, SARS-CoV-2). The ASL is an integral part of the airway defense system but not impermeable. I suggest correcting this statement.

2) In lines 169-170, the authors state: “We used HBEC cells as they faithfully reproduce the features of human bronchi.” I would suggest revising this sentence. Primary cell monolayers established on permeable support under ALI condition is the closest ex vivo model to mimic conditions regarding human airway epithelia. Although, this model fails to represent the mucus hypersecretion produced by submucosal glands in CF, which is one of the hallmarks of CF disease. Also, goblet cell hyperplasia, characteristic of CF epithelia, may or may not be present in primary cell culture. It depends on sampling the bronchial epithelium at the establishment of primary cell culture.

33)  In line 251, Trikafta contains a typo.

44)      Please change the unit of conductance to microS (μS/cm2) on the vertical axes in Figure S1, panels A and B.

Author Response

please, see attachment

Reviewer 4 Report

Dear Alessandra Ludovico, Oscar Moran and Debora Baroni,

thank you for the opportunity to read your manuscript on the in vitro CFTR modulator effects on HBECs.

Major comments:

The experiments are easy to comprehend and the results seem plausible. My major concern is the missing scientific benefit of your study. You confirm common knowledge about the mode of action of the three modulators that have already proven their valor as approved medication in the in vivo setting. Where you in doubt about the effect of the modulators on airway physiology? If yes, I am missing this initial suspicion that prompted your study in the introduction.

On the same grounds as above, please focus your introduction (which covers not only lines 34 to 70 but also 137 to 166) on the specific aim of your study. Typical questions, like where do we stand and what questions still need to be answered, are completely missing.

In your study you apply the drugs on the apical side of the ALI cultures, which is novel in a way, as the approved drugs are taken as oral medication and will thus reach the epithelia from the blood side basolaterally. Do you implement, that the drugs should be inhaled?

The introduction should explain the known mode of action of the drugs applied: correctors, potentiators and this ppq102. Why did you not use amiloride as a control condition? And it should summarize the tests used in your study and to what end they were employed.

Minor comments:

line 11: your study is on airway epithelial cells, so leave pancreas, intestine and vas deference out, as this seems like an arbitrary selection of epithelia to mention.

line 17: why not be specific and name the precise modulators you used in this study?

lines 25-26: “even better” is a very subjective and not at all scientific evaluation of the results

line 35: I would specify “genetic disease” as “monogenetic disease”

line 36: CFTR is not a cyclic nucleotide-gated channel (CNGC), as you imply. It is, however, activated by PKA-mediated phosphorylation.

lines 38-39: another arbitrary selection of epithelia to mention, different from the one in line 11 – why?

line 51: remove “could” as this is a definition of pleiotropic and nothing speculative about it.

line 56: add “a” before “great deal”

line 58: “recover the underlying defects” is probably not what you meant. Maybe “counteract”?

line 58: cumulated in not with?

figure 1 and all the following: how did you determine which concentrations of the drugs to use in your study?

lines 87-92: Why document the total protein content, when concentrations are much more meaningful? I am missing the raw data here: how much volume of fluid was exactly aspirated from each culture well and how high was the protein concentration therein?

line 96: “in vitro” is unnecessary here, as all your results come from in vitro assays.

lines 109-118: repeat lines 99-108 and can be deleted

lines 137-166: this belongs in the introductory part of the manuscript and is much better, than the original introduction.

lines 173-182: why did you not include the G=1/TEER-measurements in the main text? They seem to be as valid and important as any other results presented in the main article.

lines 185-187: Throughout the article you sometimes use the Vertex-number-system, sometimes the trivial names of the drugs, this can be very confusing. Lumacaftor and ivacaftor are mostly named as such, while tezacaftor and elexacaftor are more often referred to as VX661 and VX445.

lines 1931-195: beginning with “, as this molecule...” this needs to be part of the introduction and needs a citation to back it up.

lines 201-209: mucins are known to make water very viscous. So it is no surprise, that a higher mucin concentration renders the fluid more viscous, as you showed with your micro-rheology experiments. There needs to be no difference in quality of the mucins in the higher concentrated fluid to explain your results.

lines 217-218: “In contrast, in CF epithelia, ...the inhibitor ppq102 had no effect.” Where is the contrast? It did not have an effect on non-CF-epithelia, either.

line 223: remove “should”

lines 222-234: this is the only experiment shown, where ppq102 has an effect – and here the inhibitor is not even discussed.

lines 235-238: none of the named effects are shown in your manuscript.

line 305: Brownian motion (n)

lines 322-324: I as a reader would very much appreciate to be told exactly, what you define as replicates and as a poll under each experimental condition. Like how many different cell cultures from different donors were used? How many wells were analyzed for each experimental condition?

S1 should be included in the main article.

Author Response

please, see attachment

Round 2

Reviewer 4 Report

Dear Alessandra Ludovico, Oscar Moran and Debora Baroni,

thank you for your reply and the changes in the article, which do make it easier to read and understand.

I still have troubles understanding the statistics, though. If you have two donor cell-lines each and replicates under each condition (5 to 12) were successively pooled, than you should always have an n=2. This does not allow for any statistical analysis.

And I still do not understand, why you want to rescue or restore the defects of the del508F-CFTR. Wouldn't it be better to resore (near-)normal functioning of the mutant CFTR, instead of restoring its defects?

And I still do not understand, how you can draw conclusions on the quality of the mucins from he fluid amount and protein concentration on the apical side of your HBEC cultures after 48h of treatments. did you test the sensitivity of the assays (esp. the protein concentration measurements) with mucines in the expected concentration ranges before? Which differences would you expect and is your test specific and sensitive enough? 

Otherwise I still do not feel that your experiments add a lot to common knowledge, but if this is a major concern is for the editors to decide.

Author Response

Please, see attachment

Round 3

Reviewer 4 Report

Thank you for the minor changes.

I will leave it to the editors to decide.

Author Response

We thank the reviewer for the helpfull suggestions and comments made along the revision process. We hope that the new version of the manuscript will satisfy the requests and the expectations of the reviewer, of the editors and of the readers.